

# PCK1 as a potential hub gene in distinguishing lactate metabolism between rheumatoid arthritis and osteoarthritis

Pengfei Xin[1,2,*], Shaoqiang Pei[1,2,*], Nanshan Ma[1,2] and Lianbo Xiao[1,3]

[1] Guanghua Hospital Affiliated to Shanghai University of Traditional Chinese Medicine, Shanghai, China
[2] The Research Institute for Joint Diseases, Shanghai Academy of Traditional Chinese Medicine, Shanghai, China
[3] Department of Orthopedics, Shanghai Guanghua Hospital of Integrative Medicine, Shanghai, China
[*] These authors contributed equally to this work.

Corresponding authors
Nanshan Ma, Krystalmns@163.com
Lianbo Xiao, xiao_lianbo@163.com

## ABSTRACT

**Background**. Lactate is notably involved in the advancement of rheumatoid arthritis (RA) and osteoarthritis (OA). Nevertheless, the causal association between these conditions and lactate remains uncertain. This study aims to use Mendelian randomization (MR) to investigate their relationship with lactate and understand the genetic differences in lactate metabolism between them.

**Methods**. Genetic data for RA, OA, and lactate metabolism were obtained from GWAS, GEO, and MSigDB databases. MR analysis was performed using the inverse variance weighted (IVW) method. Differential gene expression analysis was conducted using the "limma" package, and Gene Set Enrichment Analysis (GSEA) was performed with GSEA software. Immune cell infiltration was assessed using the CIBERSORT platform. Validation of differentially expressed genes was carried out *via* Western blotting. Additionally, weighted gene co-expression network analysis (WGCNA) was employed to identify hub genes, while GO and KEGG analyses were performed to compare mechanistic differences between RA and OA. *In vitro* experiments were conducted to assess the effects of PCK1 on lactate secretion and cellular functions in RA-FLS.

**Results**. MR analysis indicated a causal relationship between RA and OA with lactate levels. Differential gene expression analysis revealed that PCK1 is a key gene underlying the metabolic differences in lactate levels between RA and OA. *In vitro* experiments demonstrated that knocking down PCK1 in RA-FLS affected lactate secretion, inhibited cell migration, and promoted apoptosis, suggesting its critical role in lactate metabolism. Additionally, GSEA analysis showed significant enrichment of PCK1 in the citrate cycle and gluconeogenesis signaling pathways in RA.

**Conclusion**. This study provides genetic evidence supporting the causal relationship between RA, OA, and lactate levels. Additionally, PCK1 is identified as a pivotal target implicated in the metabolic disparities of lactate between RA and OA, highlighting its potential significance in RA therapeutics.

## INTRODUCTION

Rheumatoid arthritis (RA) and osteoarthritis (OA) are the most prevalent forms of arthritis in clinical practice, sharing common disease features such as joint pain, stiffness, articular cartilage destruction, monocyte infiltration, inflammation, and synovial swelling (*Heo et al., 2025*; *Tang, 2019*). It is estimated that 0.5%–1% of adults in the United States suffer from RA, and its progression can potentially lead to joint deformity, severe disability, and even premature death (*Barker & Puckett, 2010*; *Di Matteo, Bathon & Emery, 2023*). Similarly, OA is the most prevalent chronic degenerative arthritis and a leading cause of chronic disability among older adults (*Palumbo et al., 2023*). RA and OA exhibit similar clinical symptoms, yet their pathological mechanisms differ significantly (*Di Matteo, Bathon & Emery, 2023*; *Ge et al., 2021*; *Heo et al., 2025*). OA is a whole-joint disease that affects multiple joint tissues, including subchondral bone remodeling, meniscal degeneration, synovial inflammation, and pathological alterations of the infrapatellar fat pad (*Cao et al., 2024*; *Fontanella et al., 2022*). In contrast, RA is characterized by immune-mediated synovial inflammation and bone destruction (*Park et al., 2025*). However, the precise mechanistic differences between the two conditions remain incompletely understood, necessitating further research for clarification.

Recent research has underscored a profound link between lactate metabolism and the pathological progression of RA (*Gan et al., 2024*; *Pucino et al., 2023*; *Yi et al., 2022*). Lactate, as the primary product of the glycolytic pathway, serves as a catalyst for the progression of RA (*Yi et al., 2022*). Multiple investigations have noted a phenomenon of low glucose and high lactate levels in the synovial fluid of RA patients, alongside a marked elevation in the local lactate/glucose ratio within RA synovial tissues (*Ahn et al., 2016*; *Quinonez-Flores, Gonzalez-Chavez & Pacheco-Tena, 2016*). Synovial lactate production can exert influences on T cells, macrophages, and osteoclasts, precipitating the progression of bone erosion in RA (*Gan et al., 2024*; *Yi et al., 2022*). Notably, it has been confirmed that hexokinase 2 (HK2), a critical enzyme in lactate metabolism, exhibits specific expression in RA. Overexpression of HK2 has been associated with heightened migratory and invasive capacities of synovial cells, along with an escalation in extracellular lactate levels (*Torres et al., 2023*). In OA research, it has also been confirmed that lactate can damage the cartilage of OA rats and exacerbate the progression of OA (*Huang et al., 2023*), while lactate synthesis inhibitors can alleviate inflammation, cartilage degradation, and chondrocyte apoptosis in OA rats (*Wen et al., 2023*). Currently, the causal relationship between RA, OA, and lactate remains unclear. In addition, the differences in lactate metabolism between the synovium of RA and OA have not been thoroughly studied. Therefore, it is necessary to explore these differences in order to provide new insights into the mechanistic differences between the two conditions and the treatment of RA.

Mendelian randomization (MR) analysis has emerged as a highly regarded method in recent years for inferring potential causal relationships. The core principle of MR is to use genetic variants, such as single nucleotide polymorphisms (SNPs), that are strongly associated with an exposure as instrumental variables (IVs) to estimate their potential causal effect on an outcome. MR simulates the design logic of randomized controlled trials

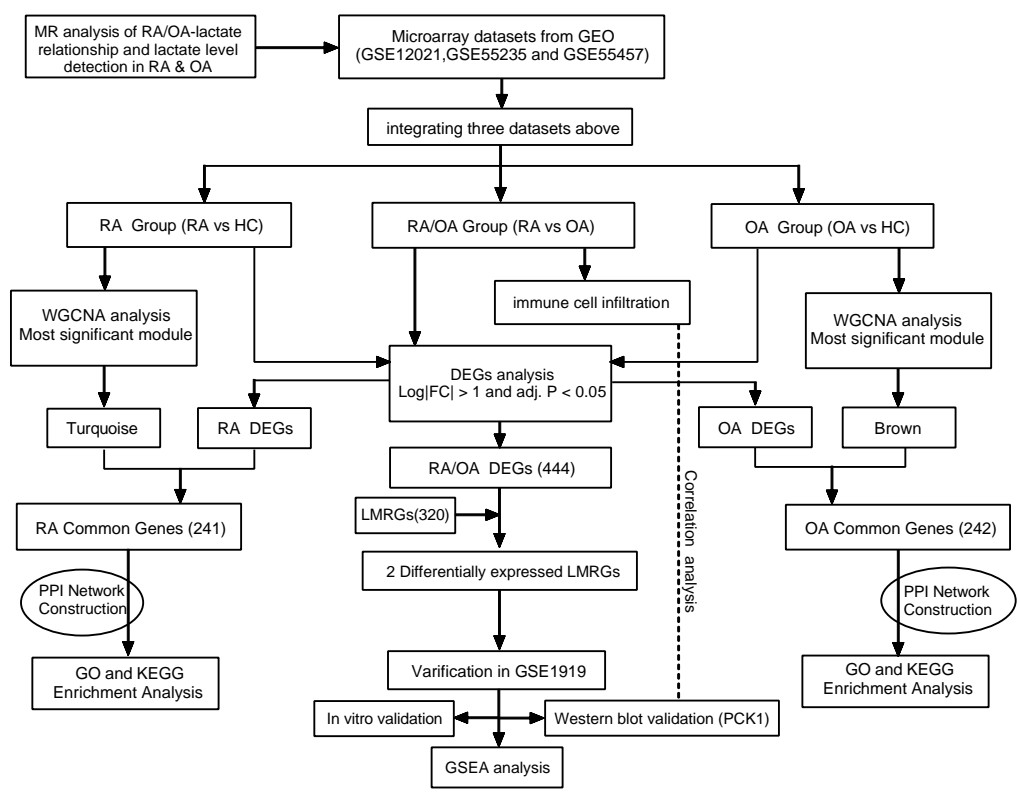

**Figure 1** **Research workflow chart.**

when assessing the causal relationship between exposure factors and outcome indicators (*Larsson, Butterworth & Burgess, 2023*; *Skrivankova et al., 2021*). To date, no studies have explored the causal relationship between RA, OA, and lactate from the perspective of genetic susceptibility. Furthermore, as bioinformatics technologies advance, microarrays on high-throughput platforms have emerged as potent instruments for discovering novel genes, biomarkers, and delving into the molecular mechanisms of diverse diseases (*Shen-Orr & Gaujoux, 2013*; *Zhao et al., 2021*).

Thus, this study aims to elucidate the causal relationships between RA, OA, and lactate through MR analysis. Furthermore, we will retrieve gene expression profiling data for RA and OA from the NCBI Gene Expression Omnibus (GEO) repository and categorize them into three groups: (1) RA/OA group (RA *vs.* OA), (2) RA group (RA *vs.* normal), and (3) OA group (OA *vs.* normal). Accessing the GSEA database can yield lactate metabolism-related genes (LMRGs). Ultimately, our goal is to pinpoint the crucial genes that underlie the metabolic differences in lactate between RA and OA, and to compare the mechanistic disparities between these two conditions. The workflow of this study is depicted in Fig. 1.

**Table 1** All data sources used for the MR analysis.

| GWAS ID | Year | Detailed information | Population | Attribute |
|---|---|---|---|---|
| ebi-a-GCST90018910 | 2021 | 8,255 cases and 409,001 controls | European | RA-Exposure |
| met-c-894 | 2016 | 24,871 Case Samples | European | RA-Outcome |
| ukb-b-14486 | 2018 | 38,472 cases and 424,461 controls | European | OA-Exposure |
| met-a-310 | 2014 | 7,814 Case Samples | European | OA-Outcome |

## MATERIALS & METHODS

### Mendelian randomization

In this study, we used a two-sample MR approach to explore the causal relationship between RA, OA, and lactate levels, with SNPs serving as IVs. RA and OA were selected as the exposure factors, and lactate levels were the outcome measure of this study. The GWAS data for RA were obtained from a meta-analysis conducted by *Sakaue et al. (2021)*, encompassing 417,256 participants of European descent. GWAS data for OA were obtained from the UK Biobank (UKB), encompassing 462,933 participants. GWAS data for lactate were sourced from metabolomics studies involving 24,871 and 7,814 participants, respectively (*Kettunen et al., 2016*; *Shin et al., 2014*). The detailed data sources are listed in Table 1.

MR analysis was conducted using the ''TwoSampleMR'' package in R (version 4.2.1) to investigate the potential causal effect of RA and OA on lactate levels. The inverse variance weighted (IVW) method served as the primary analytical approach. To ensure valid MR analysis, IVs were required to: (1) be strongly associated with the exposure (RA or OA); (2) be independent of confounders; and (3) affect the outcome (lactate levels) only *via* the exposure. Accordingly, we applied rigorous filtering steps to ensure IV quality. SNPs associated with RA or OA at a genome-wide suggestive significance level ($P < 5 \times 10^{-6}$) were initially extracted as candidate IVs. To obtain independent variants, linkage disequilibrium (LD) clumping was performed using a reference panel from the 1000 Genomes Project (European population), with an LD threshold of $R^2 < 0.001$ and a window size of 10,000 kb. Palindromic SNPs with ambiguous strand orientation were excluded to minimize bias, and weak instruments were further removed by retaining only SNPs with $F$-statistics >10. Finally, MR-Egger regression was performed to evaluate potential horizontal pleiotropy (*Emdin, Khera & Kathiresan, 2017*).

### Datasets collection

Human synovial tissue gene expression profile datasets were downloaded from the GEO database (http://www.ncbi.nlm.nih.gov/geo/) using the keywords ''rheumatoid arthritis'' and ''osteoarthritis'' (*Clough & Barrett, 2016*). The inclusion criteria for the test set are as follows: (1) expression profiling by array; (2) The datasets contain synovial tissues of patients with healthy control, OA, and RA from the joint; and (3) 30 or more synovial samples in the dataset. Finally, the three test datasets, GSE55235, GSE12021, and GSE55457, were downloaded from the GEO database, comprising a total of 94 samples; the above three datasets were all gene expression arrays and based on GPL96
**Table 2 Detailed information of selected datasets.**

| Dataset | Platform | Detailed information | Data type | Source Tissue | Attribute |
|---|---|---|---|---|---|
| GSE12021 | GPL96 | 9 normal, 10 OA, and 12 RA samples | Array | Synovium | Test set |
| GSE55235 | GPL96 | 10 normal, 10 OA, and 10 RA samples | Array | Synovium | Test set |
| GSE55457 | GPL96 | 10 normal, 10 OA, and 13 RA samples | Array | Synovium | Test set |
| GSE1919 | GPL91 | 5 OA and 5 RA samples | Array | Synovium | Validation set |
| LMRGs | MSigDB | 320 LMRGs | NA | NA | Test set |

Notes.
GPL96 refers to the Affymetrix Human Genome U133A Array; GPL91 refers to the Affymetrix Human Genome U95A Array.
LMRGs, Lactate metabolism-related genes; MSigDB, Molecular Signatures Database.

platforms. Additionally, Using the keyword "Lactate" for searching in the MSigDB (https://www.gsea-msigdb.org/gsea/msigdb) (*Liberzon et al., 2011*), we identified a total of 320 LMRGs. The GSE1919 dataset was used to validate differentially expressed LMRGs between OA and RA. The details of all data are shown in Table 2.

## Data processing and identification of DEGs

The three raw datasets were consolidated, and the "affy" package in R software was utilized for background calibration, normalization, and addressing other unwanted variations (*Irizarry et al., 2003*). When multiple probes corresponded to a common gene, the average value was calculated as its expression value. Additionally, the "sva" package was utilized to mitigate batch effects (*Leek et al., 2012*). Subsequently, differentially expressed genes (DEGs) were identified in the three datasets by comparing the gene expression profiles of synovial tissues using the "limma" package (*Ritchie et al., 2015*). To account for multiple testing, the Benjamini–Hochberg method was used to adjust $p$-values. Statistical significance was defined as an adjusted $p$-value (FDR) <0.05 and |log2 Fold change (FC)|>1.

## Differentially expressed LMRGs selection and validation

In the RA/OA group, the intersection of DEGs and LMRGs yielded differentially expressed LMRGs. Subsequently, these differentially expressed LMRGs were validated using the GSE1919 dataset.

## Ethical approval and Western blot validation

All procedures involving human participants were approved by the Ethics Committee of Shanghai Guanghua Hospital of Integrative Medicine (Approval No.: 2018-K-12). Written informed consent was obtained from all subjects prior to participation. Synovial tissues (STs) were collected from the suprapatellar region during total knee arthroplasty (TKA) in RA and OA patients treated at Shanghai Guanghua Hospital of Integrative Medicine. Inclusion criteria included clinical and radiological confirmation of RA or OA diagnosis; patients with prior immunosuppressive treatment, joint infection, or other systemic inflammatory diseases were excluded.

Tissues were homogenized in RIPA lysis buffer (#P0013B; Beyotime, Jiangsu, China) supplemented with protease inhibitors, followed by centrifugation at 12,000 rpm for 15 min at 4 °C. Protein concentrations were determined using the BCA assay (#P0010; Jiangsu,

China). Equal amounts of total protein (20 μg) were separated using 4–20% gradient precast SDS-PAGE gels (#180-9110H; Tanon, Shanhai, China) and subsequently transferred onto PVDF membranes. The membranes were probed with primary antibodies against PCK1 (1:3000, #16754-1-AP, Proteintech, Rosemont, IL, USA) and $\beta$-actin (1:1000, #66009-1-Ig, Proteintech, Rosemont, IL, USA) as a loading control. HRP-conjugated secondary antibodies (anti-rabbit IgG, #SA00001-2; anti-mouse IgG, #SA00001-1, Proteintech, Rosemont, IL, USA) were used at 1:5000 dilution. Protein bands were visualized by enhanced chemiluminescence and quantified using ImageJ software (version 1.53a).

## Fibroblast-like synoviocytes culture and grouping

STs from RA patients were processed under sterile conditions. The tissues were minced into small fragments, washed with PBS (#10010049; Gibco, Waltham, MA, USA), and digested with 100 U/mL type II collagenase (#17101015; Gibco, Waltham, MA, USA) at 37 °C for 1 h to isolate synovial cells. Cells were cultured in DMEM (#2764635; Gibco, Waltham, MA, USA) supplemented with 10% fetal bovine serum (FBS, Gibco, #10091148) at 37 °C in a 5% $CO_2$ atmosphere. When confluency reached 80–90%, cells were passaged using trypsin (#25300062; Gibco, Waltham, MA, USA). Cells from passages 3 to 6 were used in experiments.

To investigate the role of PCK1, FLS were transfected with either small interfering RNA targeting PCK1 (Si-PCK1) or a negative control siRNA (Si-NC), both purchased from GenePharma (Shanghai, China). Transfection was carried out using Lipofectamine 2000 reagent (#11668030; Invitrogen, Waltham, MA, USA) following the manufacturer's instructions. The siRNA sequences are listed in Table S1. Cells were harvested 48 h post-transfection for subsequent assays.

## Quantitative real-time PCR

Total RNA was extracted from tissue samples using the Simply P Total RNA Extraction Kit (BioFlux, China) following the manufacturer's protocol. RNA purity and concentration were determined using a NanoDrop 2000 spectrophotometer (Thermo Fisher Scientific, Waltham, MA, USA), with an A260/A280 ratio between 1.8 and 2.0 indicating high purity. Reverse transcription was performed using the PrimeScript™ RT Reagent Kit (Perfect Real Time; Takara, Shiga, Japan) to synthesize cDNA. The reaction mixture (20 μL) contained 2 μg of total RNA, 4 μL of 5x Reaction Buffer, 1 μL of PrimeScript RT Enzyme Mix, and nuclease-free water. The reaction conditions were as follows: 25 °C for 5 min (primer annealing), 50 °C for 15 min (reverse transcription), and 85 °C for 5 s (enzyme inactivation). The resulting cDNA was diluted 5–20 times with RNase-free water and stored at −20 °C for subsequent qPCR analysis.

Quantitative real-time PCR (qRT-PCR) was performed using TB Green® Premix Ex Taq™ (Tli RNaseH Plus; Takara, Shiga, Japan) on a CFX96 Real-Time PCR Detection System (Bio-Rad, Hercules, CA, USA). Each 20 μL reaction mixture contained 3 μL of diluted cDNA, 5 μL of 2x TB Green Premix, 1 μL each of forward and reverse primers (10 μM), and nuclease-free water. The thermal cycling conditions were as follows: initial denaturation at 95 °C for 1 min, followed by 40 cycles of denaturation at 95 °C for 20 s,

annealing at 55 °C for 20 s, and extension at 72 °C for 30 s. Amplification specificity was confirmed by melting curve analysis, and no-template controls (NTCs) were included to rule out contamination.

The relative mRNA expression levels were normalized to the reference gene GAPDH and calculated using the $2^{-\Delta\Delta Ct}$ method. Statistical significance was assessed using the $t$-test, with a $P$-value <0.05 considered statistically significant. Primer sequences are listed in Table S2.

## Quantification of lactate levels

Lactate levels in the cell culture supernatant were measured using lactate assay kits (BC2235, Solarbio, China) according to the manufacturer's instructions.

## Wound healing assay

RA-FLSs were seeded at a density of $1 \times 10^5$ cells per well in six-well plates and cultured until reaching over 80% confluence. A straight scratch was created on the cell monolayer using a 200 μL pipette tip, followed by washing with culture medium to remove detached cells and debris. Wound healing was monitored at 0 and 24 h post-scratch using an inverted optical microscope (Leica, DMI3000), and the scratch closure was quantified using ImageJ software.

## Flow cytometry

After collection, Cells were washed with PBS and resuspended in binding buffer at a density of $1 \times 10^6$ cells per sample. Apoptotic cells were stained using the Annexin V-FITC Apoptosis Detection Kit (Beyotime, Beijing, China) following the manufacturer's instructions. Flow cytometry analysis was performed using a CytoFLEX S flow cytometer (Beckman Coulter, Brea, CA, USA), and data were analyzed with CytExpert software.

## Evaluation of immune cell infiltration

The immune microenvironment of the RA/OA group was evaluated using CIBERSORT (https://cibersort.stanford.edu/), an online analysis tool commonly used to assess the relative content of 22 immune cell types. The proportions of these immune cells were calculated in every sample with a $P$-value <0.05 as a filter criterion and visualized in R software as a violin plot with the "vioplot" package.

## Gene set enrichment analysis

To investigate the potential molecular mechanisms of PCK1 in the occurrence and advancement of RA, we utilized GSEA to assess if predefined gene sets displayed noteworthy statistical variances between cohorts with high and low PCK1 expression levels (*Mootha et al., 2003*; *Subramanian et al., 2005*). Gene sets exhibiting a nominal (NOM) $P$-value <0.05, normalized enrichment score (NES) >1.0, and a false discovery rate (FDR) $q$-value <0.25 were considered significantly enriched.

## WGCNA construction and identification of modules

We utilized the systems biology approach WGCNA to investigate key gene modules significantly linked to the disease in both the OA and RA groups (*Langfelder & Horvath,*

*2008*). In brief, the topological overlap matrix (TOM) was employed to determine the connectivity and dissimilarity of the co-expression network established with the appropriate soft thresholding power $\beta$ (ranged from 1 to 20), and the corresponding dissimilarity (1-TOM) was calculated. Subsequently, hierarchical clustering and dynamic tree cut function were used to detect coexpressed gene modules. Finally, Pearson's test was applied to calculate the modules' correlation with clinical attributes, module membership (MM), and gene significance (GS). Modules with a *p*-value <0.05 were considered significantly related to the disease, and the eigengene network was visualized. The intersection of the DEGs identified with the integrated dataset and the modules genes most related to the disease are common genes and displayed in the form of a Venn.

### Enrichment analysis and PPI network construction

For a comprehensive grasp of the biological functions and pathways linked with common genes in both OA and RA, we performed enrichment analyses utilizing Gene Ontology (GO) and Kyoto Encyclopedia of Genes and Genomes (KEGG). Metscape, a web portal offering extensive gene list annotation and analysis resources for experimental biologists, was employed for this purpose (*Zhou et al., 2019*).

Subsequently, the protein–protein interaction (PPI) network was established using the Search Tool for the Retrieval of Interacting Genes database (STRING), with a confidence score >0.4 as the filtering criterion. The resulting PPI network was visualized using Cytoscape software (version 3.7.2). Additionally, the MCODE plugin in Cytoscape was utilized to identify significant gene modules, employing the following parameters: maximum depth = 100, node score cut-off = 0.2, degree cut-off = 2, and k-score = 2.

### Statistical analysis

Statistical methods were chosen based on data normality. Student's *t*-test was used to assess the comparative differences between two groups. Spearman correlation analysis was employed to assess the relationship between PCK1 expression and the levels of infiltrating immune cells. All analyses were conducted using SPSS software (version 24.0; IBM Corp., Armonk, NY, USA) or R software (version 4.2.1) (*R Core Team, 2018*). Correlation results were visualized using R-based charting tools. A *P*-value <0.05 was considered statistically significant.

## RESULTS

### The causal relationships between RA/OA and lactate

After strict filtering conditions, 59 SNPs of RA and 26 SNPs of OA were selected as IVs, respectively. Without of the SNPs were weak IVs, with further details regarding the characteristics of the IVs provided in Files S1 and S2. The results from the IVW method indicated an association between genetically determined RA and lactate levels (OR 1.03, 95% CI [1.00–1.05], $p = 0.021$). Similarly, OA was significantly associated with lactate levels (OR 1.45, 95% CI [1.14–1.85], $p = 0.002$) (Figs. 2A and 2B). Cochran's Q test results showed no heterogeneity between genetic predisposition to RA/OA and lactate levels (Figs. 2C and 2D). No horizontal pleiotropy was detected in the intercepts of the MR

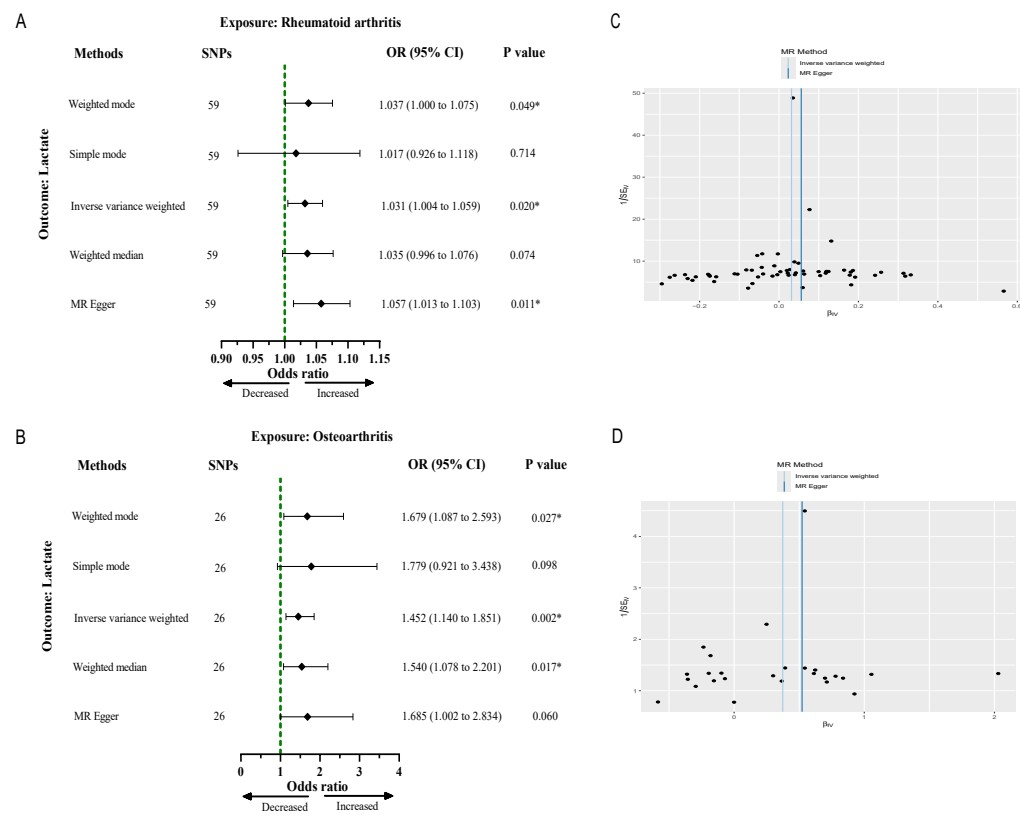

**Figure 2** Causal effect of RA and OA on the risk of increased lactate levels in Mendelian randomization analysis. (A) The causal effect of RA on lactate. (B) The causal effect of OA on lactate. (C, D) Funnel plots representing the overall heterogeneity of Mendelian randomization estimates for the effect of RA and OA on lactate, respectively. CI, confidence interval; ***$p < 0.001$.

Egger regression (RA, $p = 0.151$; OA, $p = 0.532$). The "leave-one-out" analysis revealed no individual SNP driving the overall result in the opposite direction (Figs. S1 and S2). Therefore, our results were robust and reliable.

To further clarify the differences in lactate metabolism between RA and OA, we conducted further explorations to identify differentially expressed LMRGs in RA and OA.

## DEGs screening

We conducted DEG analysis on the three groups separately. In the RA/OA group, which consisted of 35 RA and 30 OA samples, revealed 444 DEGs, including 245 upregulated and 199 downregulated genes. Within the RA group, which included 35 RA and 29 normal samples, 624 DEGs were detected, comprising 366 upregulated and 258 downregulated genes. The OA group, encompassing 30 OA and 29 normal samples, 457 DEGs were identified, with 218 upregulated and 239 downregulated genes. The volcano plots illustrating the DEGs for each group are presented in Figs. 3A–3C. To clarify the differences in LMRGs between RA and OA, we selected the DEGs from the RA/OA group for further analysis.

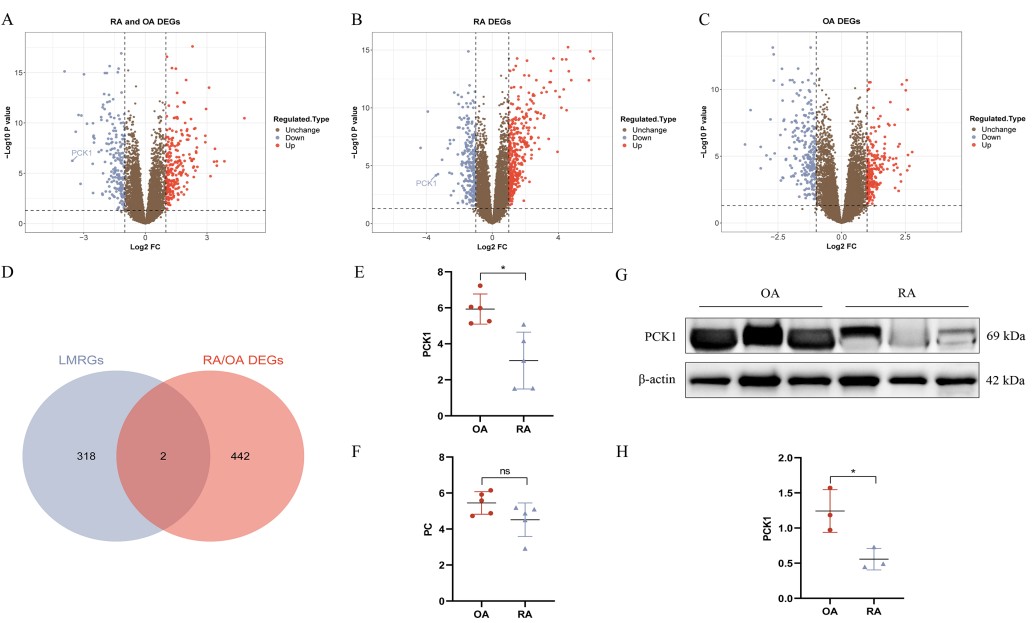

**Figure 3** **RA and OA lactate metabolism-related genes (LMRGs) selection and validation.** (A–C) Represent DEGs in RA/OA group, RA group, and OA group respectively. (D) The Venn diagram of differentially expressed LMRGs between RA and OA. (E, F) Validate the expression of PC and PCK1 in the GSE1919 dataset. (G, H) Protein expression of PCK1 in synovial tissues of both RA and OA was assessed using Western blot analysis. $*p < 0.05$, ns, no significant difference.

## Hub differentially expressed LMRGs selection and validation

In the RA/OA group, the intersection of LMRGs and DEGs revealed two common genes, specifically pyruvate carboxylase (PC) and phosphoenolpyruvate carboxykinase 1 (PCK1), as shown in Fig. 3D. Further validation of these two differentially expressed genes was conducted using the GSE1919 dataset, indicating a significant downregulation of PCK1 in the RA group ($P < 0.05$, Figs. 3E and 3F). Similarly, Western blotting results from six knee joint tissue samples showed a significant decrease in PCK1 protein expression in RA synovial tissue ($P < 0.05$, Figs. 3G and 3H).

## Downregulation of PCK1 expression affects lactate secretion in RA-FLS and influences cell function

FLS are the main cellular components of synovial tissue and play a crucial role in the progression of RA (*Meng et al., 2024*). To further elucidate the role of PCK1 in RA, we employed siRNA interference to inhibit PCK1 expression in FLS. After 48 h, significant suppression of Si-PCK1#3 expression was observed compared to the control group (Fig. 4A). Lactate level measurements revealed that Si-PCK1#3 reduced the lactate concentration in FLS supernatants, indicating that PCK1 influences lactate secretion in RA-FLS (Fig. 4B). Furthermore, scratch assays demonstrated that Si-PCK1 inhibited the migration of RA-FLS (Figs. 4C and 4D), while flow cytometry analyses indicated that Si-PCK1 promoted apoptosis in RA-FLS (Figs. 4E and 4F), consistent with previous reports on PCK1 knockout in Crohn's disease (*Yang et al., 2024*). These findings indicate

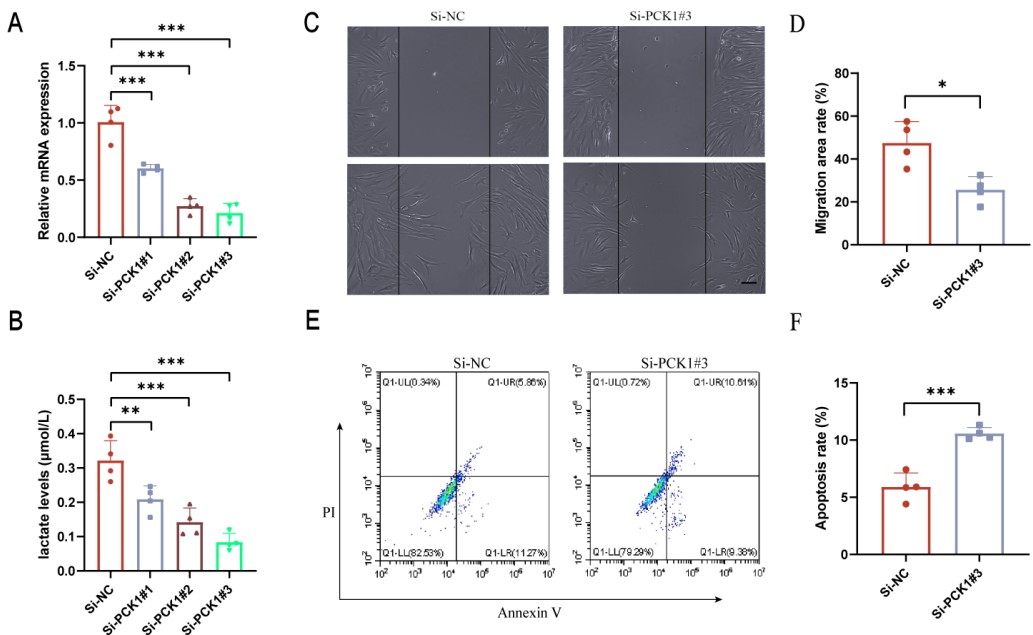

**Figure 4 The impact of downregulating PCK1 expression in RA-FLS on lactate levels and cellular functions.** (A) Verification of the knockdown efficiency of three siRNAs targeting PCK1 in RA-FLS using qPCR. (B) The effect of PCK1 knockdown on lactate metabolism in RA-FLS. (C, D) The influence of PCK1 knockdown on the migratory capacity of RA-FLS. (E, F) The effect of PCK1 knockdown on apoptosis in RA-FLS. Scale bar = 200 μm. $*p < 0.05$, $**p < 0.01$, $***p < 0.001$.

that PCK1 can influence lactate metabolism in RA-FLS, and changes in the function of FLS may be related to alterations in lactate metabolism (*Wang et al., 2022*).

## Immune cell infiltration analysis and correlation of PCK1 with immune cells

RA is a multi-system autoimmune disease closely associated with immune cells (*Komatsu & Takayanagi, 2022*). To clarify the differences in immune cell infiltration between RA and OA, we conducted further analysis using the CIBERSORT website. The results indicated significant variations in the infiltration of various immune cell types in the synovial tissues of OA and RA. Compared to OA samples, RA synovial tissues exhibited significantly higher proportions of nine immune cell types, while five immune cell types were comparatively lower (Fig. 5A). This suggests that RA is associated with greater immune cell infiltration, which may relate to its pathological mechanisms. Furthermore, to explore the relationship between immune cell infiltration and PCK1 expression, we conducted a correlation analysis. Correlation analysis (Figs. 5B–5E) indicated that, in RA, PCK1 displayed a positive correlation with plasma cells ($R = 0.53$, $P = 0.001$), while exhibiting negative correlations with M2 macrophages ($R = -0.34$, $P = 0.043$) and memory resting CD4+ T cells ($R = -0.42$, $P = 0.013$).

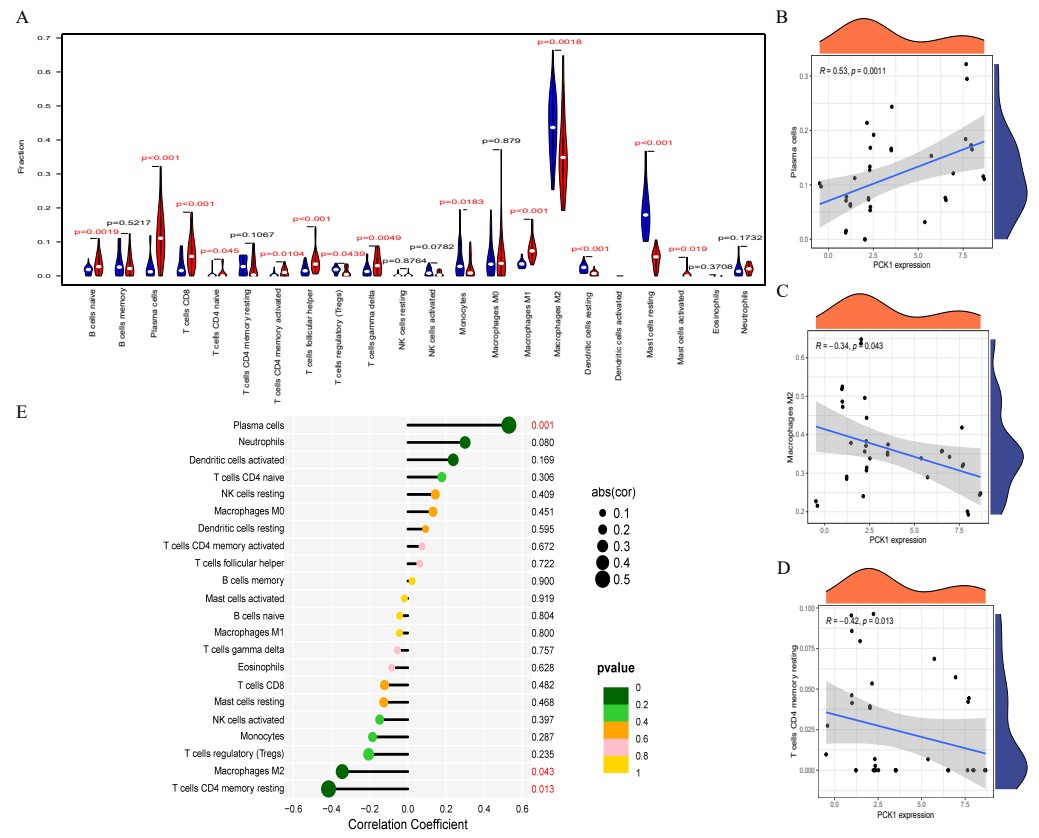

**Figure 5 The immune correlation of PCK1 in RA.** (A) The relative distribution of 22 immune cell types in synovial tissues of RA and OA. (B–E) The correlation of PCK1 with different immune cell types in RA.

## GSEA identified signaling pathways associated with PCK1

To determine the potential signaling pathways linked to the regulatory mechanism of PCK1 in RA, we performed a comparative analysis employing the GSEA method. The results demonstrated a notable correlation between the expression levels of PCK1 and 19 signaling pathways, including CITRATE_CYCLE_TCA_CYCLE (NES = 2.08, $P = 0.000$, $q = 0.005$) and GLYCOLYSIS_GLUCONEOGENESIS (NES = 1.63, $P = 0.019$, $q = 0.097$) (as shown in Figs. 6A and 6B and Table 3).

## The co-expression modules in RA and OA groups

In addition to the differences in lactate metabolism mechanisms, we also performed WGCNA analysis for the RA and OA groups to further compare the mechanistic differences between them. All samples and 12,483 genes from both groups were utilized for WGCNA analysis. In the OA group, a soft threshold $\beta$ of 8 (R2 = 0.85) was chosen, ensuring the gene association's consistency with the scale-free distribution. The threshold was set at 0.4, and the minimum number of genes in a module was set to 30 to facilitate the merging of similar modules in the cluster tree. A hierarchical clustering tree was constructed following a dynamic hybrid cut (Fig. 7A). Ultimately, 16 modules were identified within the co-expression network. The MEbrown module ($r = -0.84$, $p = 7e-17$) significantly
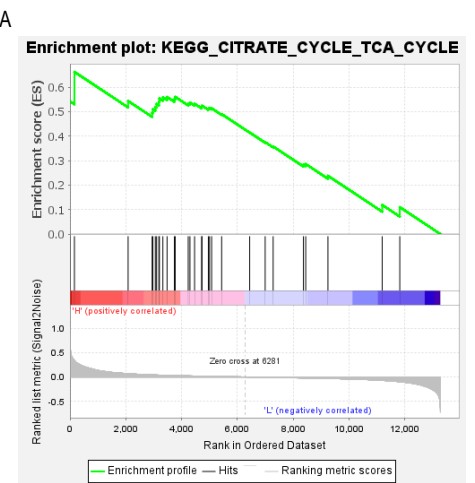
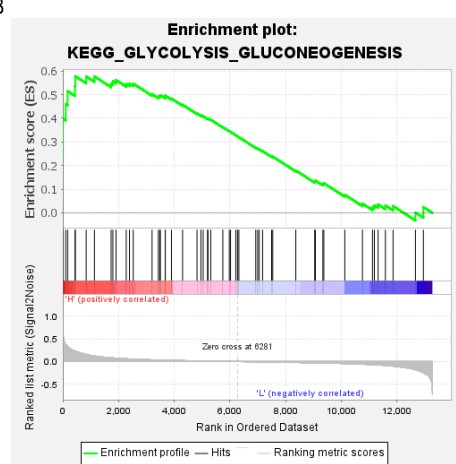

**Figure 6   GSEA enrichment plots.** The GSEA results revealed significant enrichment of genes associated with PCK1 in key signaling pathways in RA, including (A) the tricarboxylic acid cycle and (B) gluconeogenesis signaling pathway.

**Table 3   Enrichment parameters of PCK1 analyzed by GSEA.**

| Gene set name | ES | NES | NOM<br>*P*-value | FDR<br>*q*-value |
|---|---|---|---|---|
| KEGG_CITRATE_CYCLE_TCA_CYCLE | 0.66 | 2.08 | 0.000 | 0.005 |
| KEGG_ADIPOCYTOKINE_SIGNALING_PATHWAY | 0.55 | 1.90 | 0.000 | 0.019 |
| KEGG_FATTY_ACID_METABOLISM | 0.69 | 1.85 | 0.002 | 0.025 |
| KEGG_PPAR_SIGNALING_PATHWAY | 0.69 | 1.78 | 0.000 | 0.041 |
| KEGG_PYRUVATE_METABOLISM | 0.59 | 1.77 | 0.006 | 0.037 |
| KEGG_PROPANOATE_METABOLISM | 0.61 | 1.76 | 0.010 | 0.038 |
| KEGG_PROXIMAL_TUBULE_BICARBONATE_RECLAMATION | 0.76 | 1.75 | 0.000 | 0.033 |
| KEGG_GLYCOLYSIS_GLUCONEOGENESIS | 0.58 | 1.63 | 0.019 | 0.097 |
| KEGG_STEROID_BIOSYNTHESIS | 0.65 | 1.62 | 0.027 | 0.091 |
| KEGG_VALINE_LEUCINE_AND_ISOLEUCINE_DEGRADATION | 0.47 | 1.60 | 0.040 | 0.105 |
| KEGG_PEROXISOME | 0.41 | 1.59 | 0.021 | 0.097 |
| KEGG_GLUTATHIONE_METABOLISM | 0.40 | 1.56 | 0.027 | 0.121 |
| KEGG_ALANINE_ASPARTATE_AND_GLUTAMATE_METABOLISM | 0.57 | 1.55 | 0.023 | 0.115 |
| KEGG_TYROSINE_METABOLISM | 0.65 | 1.55 | 0.014 | 0.113 |
| KEGG_LYSINE_DEGRADATION | 0.47 | 1.48 | 0.036 | 0.180 |
| KEGG_RETINOL_METABOLISM | 0.57 | 1.48 | 0.008 | 0.171 |
| KEGG_BETA_ALANINE_METABOLISM | 0.57 | 1.48 | 0.045 | 0.161 |
| KEGG_DRUG_METABOLISM_CYTOCHROME_P450 | 0.55 | 1.48 | 0.027 | 0.156 |
| KEGG_TRYPTOPHAN_METABOLISM | 0.50 | 1.43 | 0.030 | 0.211 |

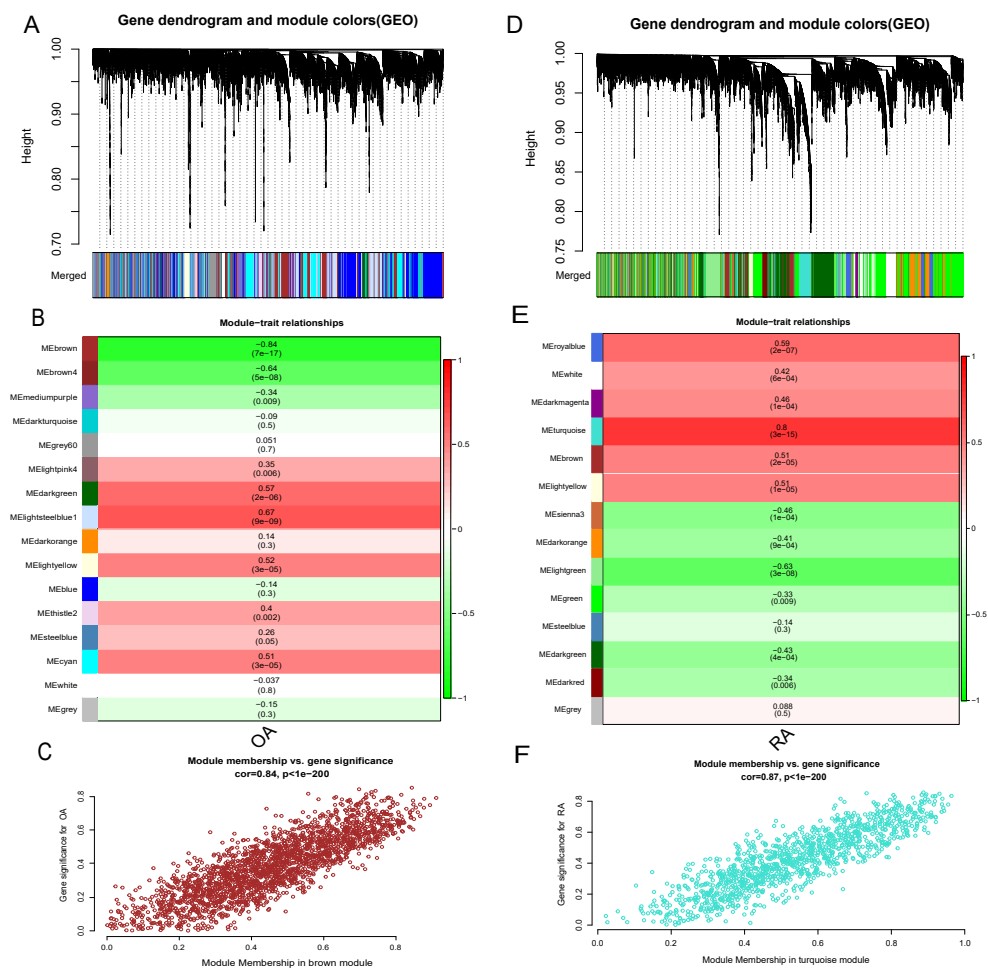

**Figure 7 Identification of gene modules associated with RA and OA in the GEO dataset using WGCNA.** (A, D) Different colors under the gene tree represent gene co-expression modules. (B, E) Module-trait relationships were assessed by correlating module eigengenes with clinical traits, with each cell indicating the corresponding correlation coefficient and associated *P* value. (C, F) Scatter plots of module eigengenes in the brown and turquoise modules.

correlated with OA (Figs. 7B and 7C). Further analysis was conducted using 2,217 genes in the brown module. Similarly, in the RA group, hierarchical clustering tree maps and heatmaps of module-trait relationships were constructed (Figs. 7D and 7E). Within the 14 modules, the MEturquoise module ($r = 0.8$, $p = 3e{-}15$) displayed the highest correlation with RA (Fig. 7F), encompassing 1,205 genes. The intersection of DEGs screened from the integrated dataset in the OA and RA groups and the genes in the brown and turquoise modules represents common genes associated with OA and RA, respectively.

## Enrichment analysis of common genes in the RA and OA groups

In the RA group, 241 intersecting genes were identified (Fig. 8A). GO enrichment analysis revealed that the common genes were mainly enriched in various biological processes

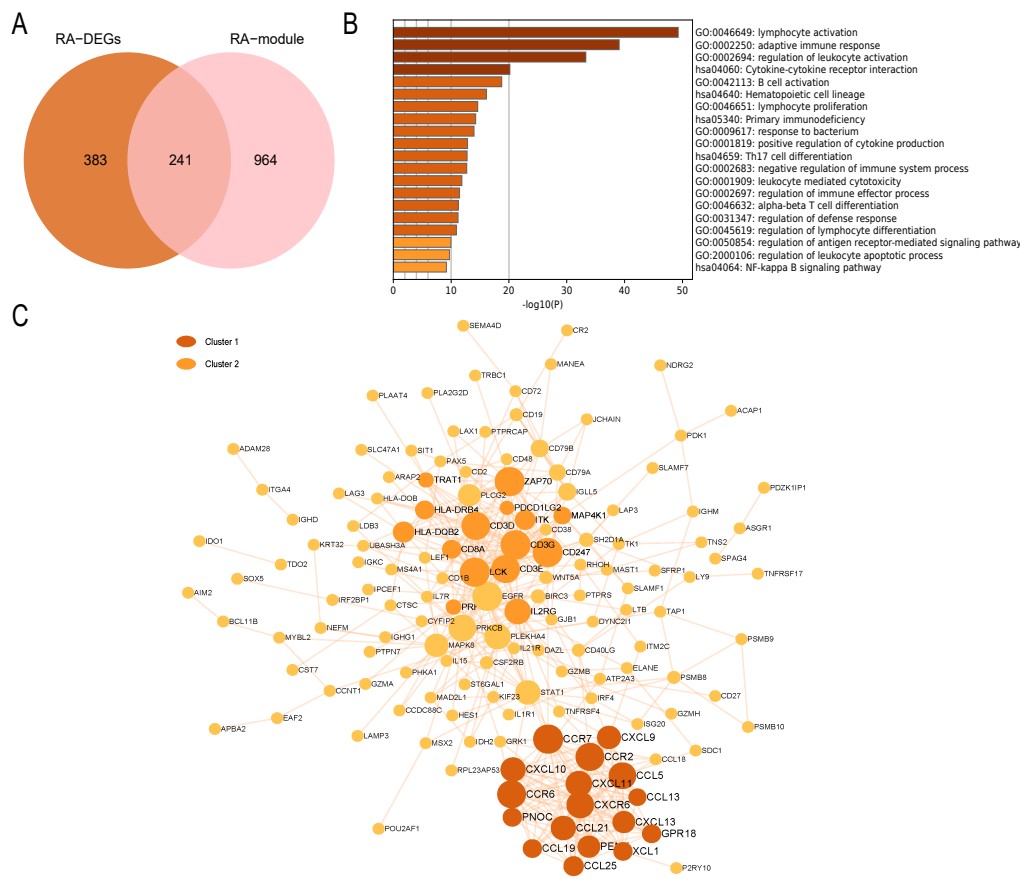

**Figure 8** **PPI network and enrichment analysis of RA common genes.** (A) The Venn diagram of the RA common genes. (B) GO and KEGG enrichment analysis of the RA common genes. (C) The PPI network of the RA common genes. Gray and yellow are Cluster 1 and 2, respectively.

and pathways, such as lymphocyte activation, adaptive immune response, regulation of leukocyte activation, B cell activation, Th17 cell differentiation pathway, and NF-kappa B signaling pathway. In summary, these biological processes and pathways are predominantly associated with immunity and inflammation, as depicted in Fig. 8B. The PPI network visualized by Cytoscape comprises 148 nodes and 452 edges. Module analysis using the MCODE plugin identified two clustered modules based on filtering criteria. Cluster 1 obtained the highest score (14.5, with 17 nodes and 116 edges), while Cluster 2 followed closely behind (with a score of 10.429, 15 nodes, and 73 edges). The genes within the cluster are enriched in lymphocyte chemotaxis and adaptive immune response, respectively (Fig. 8C).

In the OA group, we detected 241 intersecting genes (Fig. 9A). Similarly, GO enrichment analysis indicated their predominant enrichment in biological processes such as blood vessel development, transcription regulator complex, and regulation of smooth muscle cell proliferation alongside signaling pathways, including the TNF and MAPK signaling pathways, as illustrated in Fig. 9B. These processes and pathways are also correlated with

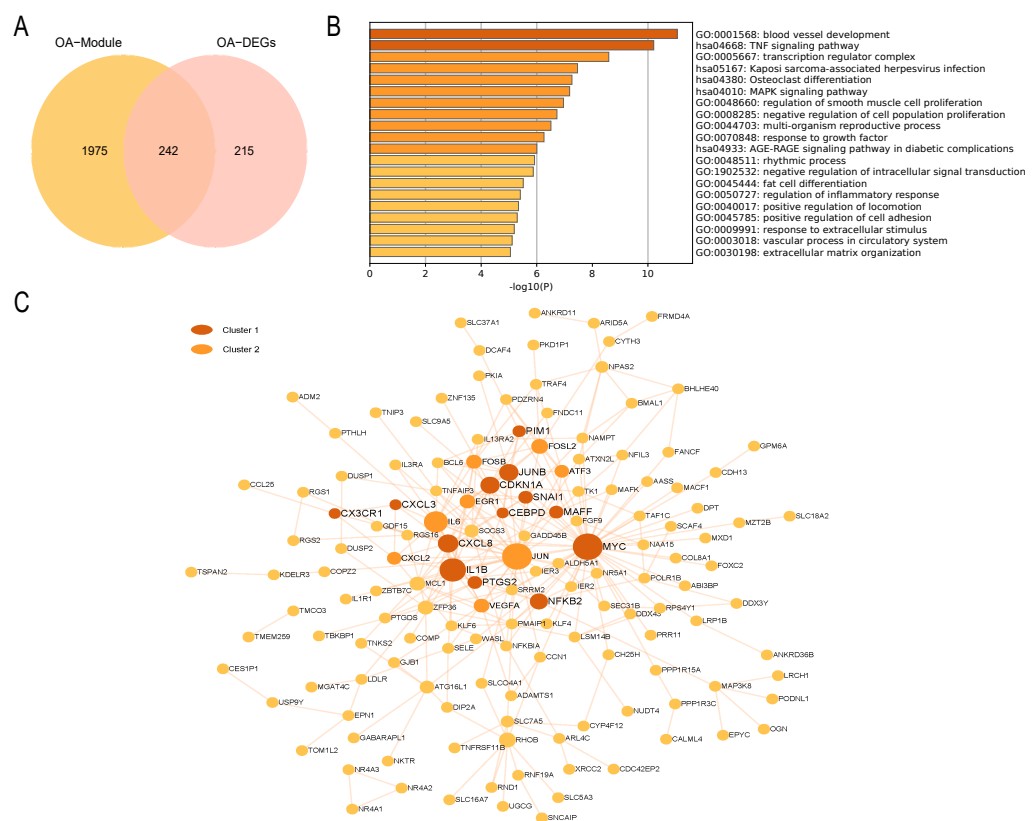

**Figure 9** **PPI network and enrichment analysis of OA common genes.** (A) The Venn diagram of the OA common genes. (B) GO and KEGG enrichment analysis of the OA common genes. (C) The PPI network of the OA common genes. Gray and yellow are Cluster 1 and 2, respectively.

inflammation. We identified a clustered module in the visualized PPI network, with Cluster 1 achieving the highest score (Score: 4.2, 11 nodes, and 21 edges). These genes are primarily enriched in Signaling by Interleukins, as detailed in Fig. 9C. In summary, both RA and OA groups show significant immune and inflammatory-related pathways, highlighting the underlying commonalities and differences in their pathophysiology.

# DISCUSSION

RA and OA represent the two most prevalent forms of arthritis encountered in clinical settings, often posing challenges in their differentiation (*Ge et al., 2021*; *Ureten & Maras, 2022*). Lactate plays a significant role in the progression of both diseases (*Huang et al., 2023*; *Yi et al., 2022*). Through MR studies, we found that genetically determined RA and OA are associated with lactate levels. Further investigations revealed that the expression of PCK1, a LMRG, is lower in RA synovial tissues compared to OA. Silencing PCK1 in RA-FLS can affect lactate secretion, leading to a reduction in cell migration ability and promoting apoptosis. GSEA indicated that high PCK1 expression in RA is significantly enriched in the CITRATE CYCLE TCA CYCLE and GLYCOLYSIS_GLUCONEOGENESIS

pathways. Therefore, we propose that PCK1 is a hub gene contributing to the differences in lactate metabolism between RA and OA.

In our previous research, we observed a notable increase in serum lactate levels among patients with RA and OA (*Jiang et al., 2013*). Additionally, multiple studies have confirmed that lactate levels are also elevated in the synovial fluid of the knee joints in RA and OA (*Ahn et al., 2016*; *Garcia-Carbonell et al., 2016*). The elevation of lactate levels may be attributed to various factors, including hypoxia in synovial tissue and enhanced glycolysis (*Bustamante et al., 2017*; *Quinonez-Flores, Gonzalez-Chavez & Pacheco-Tena, 2016*), but no study has yet confirmed a causal relationship between RA/OA and lactate levels. This study, based on large-scale GWAS data, found a causal relationship between RA/OA and lactate levels. The concept of MR is similar to that of prospective randomized controlled trials (RCTs), but it effectively avoids the influence of reverse causality and various biases. To ensure that the SNPs were unrelated to any confounding factors between RA/OA and lactate, we only selected participants from European populations. We employed the IVW model to minimize heterogeneity's influence as much as possible, ensuring all outcomes were devoid of horizontal pleiotropy. Consequently, our MR results are deemed reliable, furnishing robust genetic evidence underpinning the causal nexus between RA/OA and lactate levels.

PCK1 plays a pivotal role as a rate-limiting enzyme in gluconeogenesis, responsible for catalyzing the conversion of oxaloacetate (OAA) and GTP into phosphoenolpyruvate (PEP) and carbon dioxide by adding phosphate to pyruvate and simultaneously cleaving carbon dioxide from OAA (*Liu et al., 2024*; *Lu et al., 2023*). Recent research has yielded numerous findings on PCK1 in tumor studies, yet its role remains controversial, exhibiting both tumor-suppressive and oncogenic properties across various malignancies. Nevertheless, it is evident that PCK1 influences disease progression by modulating lactate metabolism (*Abate et al., 2023*). Our investigation revealed that PCK1 expression is significantly lower in the synovial tissues of RA compared to OA. Consistent with this, *Zhao et al. (2016)* reported a downregulation of PCK1 in the synovial tissues of CIA rats. Additionally, our *in vitro* experiments demonstrated that silencing PCK1 in RA-FLS alters lactate levels, inhibits cell migration, and promotes apoptosis. The increased lactate secretion is a hallmark of RA-FLS activation, while reduced lactate levels correlate with the functional state of RA-FLS. The proliferation and migration of FLS represent critical pathological features of RA, where rapid RA-FLS proliferation can lead to elevated lactate levels (*Bustamante et al., 2017*; *Wang et al., 2022*). Consequently, the functional alterations in FLS resulting from PCK1 knockout may be linked to changes in lactate metabolism. The pathways associated with lactate metabolism identified through GSEA analysis further substantiate our hypothesis. Similarly, *Verissimo et al. (2023)* found that PCK1 knockout induces alterations in lactate metabolism in kidney disease research. Thus, our study uncovers novel targets for distinguishing RA from OA and provides valuable insights for future research on PCK1 in RA.

Additionally, Our study entailed a comparative analysis delineating the mechanistic disparities between RA and OA. Enrichment analysis showed that genes differing in RA were mainly enriched in the NF-kappa B signaling pathway. The excessive activation

of NF-$\kappa$B is closely associated with the pathogenesis of RA, as this pathway is involved in the synovial inflammation and bone destruction in RA, promoting the onset and progression of the disease (*Clanchy et al., 2021*; *Ren et al., 2023*; *Schett & Gravallese, 2012*). Differential genes in OA were predominantly enriched in the TNF and MAPK signaling pathway. Similarly, the activation of these pathways can induce chondrocytes to synthesize MMP, proteoglycanase, and other factors, ultimately leading to joint inflammation and degradation of the cartilage matrix, thereby promoting the progression of OA (*Xu et al., 2022*; *Zhang et al., 2021*). Both the mechanisms of RA and OA are inherently linked to inflammation. However, RA exhibits a more intricate association with the functionality of immune cells, corroborating with the findings of immune cell infiltration analysis.

Despite our study yielding significant findings, several limitations remain. Firstly, the GWAS data utilized in this research primarily derive from populations of European ancestry, which restricts the generalizability of the conclusions to other ethnic groups. Secondly, although our study identified PCK1 as a participant in lactate metabolism in RA, the specific mechanisms underlying PCK1's role in RA require further investigation. Finally, our previous research indicated that synovial tissues exhibit distinct metabolic characteristics during relapsed and remission phases (*Meng et al., 2024*), and the regulatory role of PCK1 in lactate metabolism within synovial tissues with varying metabolic profiles still needs clarification.

## CONCLUSIONS

Overall, our MR study provides genetic evidence for the causal relationship between individuals with RA and OA and lactate levels. Additionally, we identified PCK1 as a key target for the differences in lactate metabolism between RA and OA. Our findings highlight a novel target that may play a significant role in the future treatment of RA.

## ACKNOWLEDGEMENTS

This work has benefited from GEO and GWAS, and I would like to express my gratitude to my girlfriend, Ms. Mengnan Ke, for her strong support of my scientific research work.

### Funding
This research was funded by the National Natural Science Foundation of China (Grant No. 82474302). The funders had no role in study design, data collection and analysis, decision to publish, or preparation of the manuscript.

### Grant Disclosures
The following grant information was disclosed by the authors:
National Natural Science Foundation of China: 82474302.

### Competing Interests
The authors declare there are no competing interests.

## Author Contributions

- Pengfei Xin conceived and designed the experiments, performed the experiments, analyzed the data, prepared figures and/or tables, authored or reviewed drafts of the article, and approved the final draft.
- Shaoqiang Pei conceived and designed the experiments, performed the experiments, prepared figures and/or tables, and approved the final draft.
- Nanshan Ma conceived and designed the experiments, analyzed the data, authored or reviewed drafts of the article, and approved the final draft.
- Lianbo Xiao analyzed the data, prepared figures and/or tables, authored or reviewed drafts of the article, and approved the final draft.

## Human Ethics

The following information was supplied relating to ethical approvals (*i.e.*, approving body and any reference numbers):

The Ethics Committee of the Shanghai Guanghua Hospital of Integrative Medicine.

## Data Availability

The data is available at Zenodo: XIN, P. (2025). The impact of PCK1 knockout on the apoptosis of FLS cells. [Data set]. Zenodo. https://doi.org/10.5281/zenodo.15654040.

The GEO datasets used in this study are publicly available from NCBI GEO: GSE12021, GSE55235, GSE55457, and GSE1919.

The GWAS summary statistics used in this study are available from the OpenGWAS database: ebi-a-GCST90018910, met-c-894, ukb-b-14486 and met-a-310.

## Supplemental Information

Supplemental information for this article can be found online at http://dx.doi.org/10.7717/peerj.19661#supplemental-information.

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
