# Peer review of "PCK1 as a potential hub gene in distinguishing lactate metabolism between rheumatoid arthritis and osteoarthritis"

_PeerJ, doi:10.7717/peerj.19661_

## Round 0.1 · original submission · Major Revisions

Reviewer 1 ·

Basic reporting

no comment

Experimental design

no comment

Validity of the findings

no comment

Additional comments

1. Please add a Conclusions section to the abstract.
2. I did not find any description of the keywords in the manuscript. May I ask if this part has been omitted? Please verify with the author.
3. Line 55-56, please modify the format.
4. The format of the primary and secondary headings in the article is not consistent.
5. "The protein expression of PCK1 was measured by standard western 136 blot assay from synovial tissues of RA and OA patients." "Synovial tissues (STs) were obtained from RA patients under knee synergy at 144 Shanghai Guanghua Hospital of Integrated Medicine." Please explain the source of synovial tissues of OA patients and suggest rewriting sections 2.5 and 2.6.
6. Figure 4, please describe the meanings of "*", "* *", and "* * *".

Reviewer 2 ·

Basic reporting

no comment

Experimental design

My major concern regarding this manuscript is that the authors should explain the rational to use MR to explore this topic. MR is appropriate to find associations between genetic components and possible phenotypes, risks, etc. Thus, the first point is to demonstrate a proven causative relationship between certain SNPs and lactate metabolism and OA/RA. The authors should provide sufficient detail and background on choosing the SNPs. Otherwise, this part should be deleted.

Methodology should be improved.

Validity of the findings

Lines 125-127: did authors use a p-value < 0.05 or the adjusted p-value? Please clarify for all bioinformatics analysis.

Additional comments

Lines 49-51: please add appropriate references. Reference 2 is old.
Lines 56-58: authors should clearly explain that OA is a whole joint disease, involving all joint tissues including meniscal degeneration and changes in the infrapatellar fat pad (DOI:10.3390/biomedicines10061369 etc).
Lines 104-107: this part should be expanded.
Line 104 and line 119: authors mentioned table 1 for both SNPs and GEO datasets. Please separate the data into two tables. Authors must explain the rationale of choosing these SNPs. GEO dataset should be better reported. Please add tissue type and type of array used.
Line 136: it seems that authors performed western blots on synovial tissues. If yes, authors should add inclusion/exclusion criteria of patient’s enrolment. Did authors collect suprapatellar synovial membrane during total knee replacement? Protein extraction protocol should be added. SDS.PAGE used ect should be reported. Please clarify.
Line 145: ml should be mL. Supplier of all reagent should be added.
Line 147: CO2 should be corrected.
Line 151: please add sequences.
Line 179: in the supernatant of the cells? Please clarify.
Lines 182, 188: how many cells were seeded?
Line 185: microscope used should be reported.
Line 190: instrument used should be added.
Lines 199-201: this part should be moved to the statistical analysis section.
Lines 234-235: SPSS should be cited according the supplier.
Lines 238-239: each SNPs used should be motivated. IVs is unclear.
Why did authors select to silence PCK1 and not PC?
Figure 4c: better images should be supplied. These are blurred.
Line 342: secretion or production?
Funding is lacking.
Please use the same font throughout the manuscript.

---

## Round 0.2 · accepted · Accept

Dear Authors,

Thank you for sending the revised version of the manuscript. Based on the Author's responses to both Reviewers' comments (and Reviewer 2's acceptance), I recommend the article for publication. Congratulations!

Please double check missing whitespaces (I think I spotted at least one) when accepting the proofs.

Reviewer 2 ·

Basic reporting

Authors replied to all my questions.

Experimental design

Authors replied to all my questions.

Validity of the findings

Authors replied to all my questions.

Additional comments

Authors replied to all my questions.